# TEST-TIME TRAINING FOR SEMANTIC SEGMENTATION WITH OUTPUT CONTRASTIVE LOSS

## ABSTRACT

Although deep learning-based segmentation models have achieved impressive performance on public benchmarks, generalizing well to unseen environments remains a major challenge. To improve the model's generalization ability to the new domain during evaluation, the test-time training (TTT) is a challenging paradigm that adapts the source-pretrained model in an online fashion. Early efforts on TTT mainly focus on the image classification task. Directly extending these methods to semantic segmentation easily experiences unstable adaption due to segmentation's inherent characteristics, such as extreme class imbalance and complex decision spaces. To stabilize the adaptation process, we introduce contrastive loss (CL), known for its capability to learn robust and generalized representations. Nevertheless, the traditional CL operates in the representation space and cannot directly enhance predictions. In this paper, we resolve this limitation by adapting the CL to the output space, employing a high temperature, and simplifying the formulation, resulting in a straightforward yet effective loss function called Output Contrastive Loss (OCL). Our comprehensive experiments validate the efficacy of our approach across diverse evaluation scenarios. Notably, our method excels even when applied to models initially pre-trained using domain adaptation methods on test domain data, showcasing its resilience and adaptability.

## 1 INTRODUCTION

Over the last few years, deep neural networks (DNNs) have exhibited impressive performance on standard benchmark datasets for semantic segmentation, such as Cityscapes and PASCAL VOC Long et al. (2015); Ronneberger et al. (2015); Chen et al. (2017); Xie et al. (2021); Strudel et al. (2021). However, DNNs often struggle to generalize under distribution shifts, and their performance degrades considerably when the training data differs from the test data. Unfortunately, in real-world scenarios such as autonomous driving, domain shifts are inevitable due to variations in weather conditions, illumination, sensor sensitivity, and so on.

Domain adaptation (DA) methods Hoffman et al. (2018); Luo et al. (2019); Yang & Soatto (2020); Tranheden et al. (2021); Zou et al. (2018); Vu et al. (2019); Hoyer et al. (2022a;b) attempt to improve model performance on the target domain in the presence of a distribution shift between the source and target domains. However, DA methods require offline model tuning using the entire set of target samples, which is impractical when collecting the entire set of target samples in advance is unfeasible or when immediate predictions are needed. To address this issue, test-time training (TTT) adapts a well-trained source model with currently available test data during evaluation. Resorting to self-supervised regularization Sun et al. (2020); Liu et al. (2021b); Gandelsman et al. (2022), entropy regularization Wang et al. (2020); Zhang et al. (2021a); Wang et al. (2022); Brahma & Rai (2023), consistency regularization Zhang et al. (2021a); Wang et al. (2022); Chen et al. (2022), diversity regularization Chen et al. (2022); Döbler et al. (2022); Jang et al. (2022), distillation regularization Döbler et al. (2022); Chen et al. (2022); Wang et al. (2022) or the combination of those terms Zhang et al. (2021a); Döbler et al. (2022); Chen et al. (2022); Wang et al. (2022), TTT methods have achieved significant progress in the field of image classification. However, recent studies Wang et al. (2022); Gao et al. (2022a); Niu et al. (2023); Lim et al. (2023) revealed that the existing TTT methods may be unstable in challenging scenarios like small test batch sizes, class imbalance, and continual domain shift.

Segmentation tasks present unique challenges when applying existing TTT methods. Unlike classification tasks, segmentation often encounters more severe class imbalance issues, which leads to biased decision boundaries towards majority classes while overlooking minority classes. Additionally, segmentation is a complex task involving structured prediction. Unlike classification, its decision function is more intricate because it must make predictions within an exponentially large label space Zhang et al. (2017); Tsai et al. (2018). These differences can easily result in error accumulation Wang et al. (2022) and even mode collapse Niu et al. (2023) when applying existing TTT methods to segmentation tasks. Both prior research Shin et al. (2022); Song et al. (2022); Wang et al. (2022) and our own experimental findings confirm that existing techniques proven may not necessarily enhance segmentation performance effectively. Furthermore, the design of the encoder-decoder architecture in segmentation tasks is more complex than the encoder-only design in classification tasks. Therefore, certain advanced techniques commonly applied in classification, such as those dependent on feature statistics estimated from a large batch of test samples Liu et al. (2021b), or those requiring additional self-supervised heads Sun et al. (2020); Liu et al. (2021b), may not be well-suited for segmentation tasks.

In pursuit of enhanced adaptation stability in the TTT for segmentation, this study introduces a novel objective derived from the principles of contrastive learning. The effectiveness of contrastive learning in developing robust and versatile representations has been well-established in both pre-training Chen et al. (2020); He et al. (2020) and adaptation Liu et al. (2021a); Chen et al. (2022); Wang et al. (2021b) scenarios. These methods typically apply the contrastive loss (CL) within the representation space, necessitating the integration of fine-tuning or additional regularization techniques to align the task head with the learned representation effectively. However, in the context of TTT for segmentation, fine-tuning with labeled data is restricted, and introducing extra regularization demands intricate adjustments and limits the method's applicability. To relieve dependence on fine-tuning or intricate regularization, we expand the utility of the CL. This expansion involves a shift from its original application in the representation space (post-encoder) to the output space (post-task-head). Moreover, choosing a high temperature is required as employing the CL on the output space to avoid too uniform class distribution. By employing an infinitely high temperature, we simplify the formulation and then use it in our task. We coin our newly proposed loss formulation as Output Contrastive Loss (OCL), with a primary focus on exploring the applicability of CL in the output space.

Our comprehensive experimental results consistently illustrate that the OCL improves the model's generalization across various architectures, task settings (including TTT and continual TTT), and benchmark datasets. Surprisingly, our method demonstrates the ability to further boost model performance when the model has been initially trained using DA techniques. Compared with the regular setting where TTT methods fine-tune a model trained on the source domain, a model trained using DA techniques has been exposed to data sampled from the target domain. Consequently, the baseline model's performance closely approaches the upper performance bound in this scenario.

## 2 RELATED WORK

**Domain adaptation for semantic segmentation.** DA is a common technique used to improve the performance of semantic segmentation models when being applied to data from new, unseen domains. The majority of popular methods in this field can be divided into three categories: discrepancy minimization, image translation, and self-training. Discrepancy minimization methods aim to reduce the performance drop caused by domain shift by minimizing the difference between the source and target domains. One example of this is domain adversarial training, which trains a feature encoder to generate features that are indistinguishable by a domain discriminator Vu et al. (2019); Hoffman et al. (2018). Image translation generates target-like source images for model adaptation, either through simple alteration of low-frequency components Yang & Soatto (2020) or via conditional generative adversarial learning Hoffman et al. (2018); Dou et al. (2018); Chen et al. (2017). Self-training involves iteratively generating pseudo labels for target data and refining the model with them to improve performance. To increase the robustness of the self-training, multiple regularization techniques like class-balanced sampling strategy Zou et al. (2018), confidence regularization Zou et al. (2019); Vu et al. (2019), domain-mixup Tranheden et al. (2021) and consistency regularization Hoyer et al. (2022a;b); Tranheden et al. (2021), are introduced.

DA methods inevitably require access to both the entire set of labeled source images and unlabeled target images to tune the model. Although recently proposed source free domain adaptation (SFDA) Liu et al. (2021a); Kundu et al. (2021) is able to deal with the scenario where source data are unavailable, collections of target images are indispensable for the offline model tuning. To be more applicable to real-world scenarios, this paper focuses on the TTT, a scenario that adapts the model to the incoming test data in an online fashion. Moreover, we emphasize that the TTT can also be combined with DA methods to achieve more accurate adaptation.

**Test-time training for classification.** In recent years, a variety of methods have been proposed to address the TTT problem in the field of image classification, each with distinct objectives. One line of studies mainly relies on the self-supervised loss Sun et al. (2020); Liu et al. (2021b); Gandelsman et al. (2022), which leverages the strong correlation between the main task and a self-supervised task. Specifically, the self-supervised task is used to replace the main task to adapt the model at test time, after jointly optimizing the main task and the auxiliary self-supervised task during training. Another line of work seeks and utilizes hints behind predictions. For instance, TENT Wang et al. (2020); Niu et al. (2022) employs entropy as an objective due to its connections to error and shift, while CL Liu et al. (2021b); Chen et al. (2022); Döbler et al. (2022); Gao et al. (2022b), consistency loss Zhang et al. (2021a); Wang et al. (2022); Döbler et al. (2022), and diversity loss Döbler et al. (2022); Wang et al. (2020) are also introduced. Usually, these terms are combined for the better performance. Our method shares similar insights with the prior work in devising a loss for enhancing model generalization on unlabeled test data, but we find a majority of existing methods cannot stabilize training in the context of semantic segmentation. To address this issue, we propose a simple loss formulation tailored to enhance training stability. Moreover, some methods Wang et al. (2020); Döbler et al. (2022); Boudiaf et al. (2022); Zhang et al. (2021b); Chen et al. (2022); Niu et al. (2022) assume that a large batch of test data is available to adapt the model at a time, while others Sun et al. (2020); Zhang et al. (2021a); Gao et al. (2022a); Bartler et al. (2022); Gandelsman et al. (2022); Lim et al. (2023) tackle a more challenging yet practical scenario where only a small batch of data is available. Our study follows the latter setting, as this is more practical in semantic segmentation.

**Test-time training for segmentation.** There has been a limited amount of research focused on exploring TTT for segmentation. One such approach is the MM-TTA framework Shin et al. (2022), which is designed for 3D semantic segmentation and employs multiple modalities to provide self-supervisory signals to one another. However, the requirement for multiple modalities also limits their applications to the single modality scene. Another approach, CD-TTA Song et al. (2022), also explores TTT for urban scene segmentation but differs in setting where it trains the model on a training set and then validates on a separate validation set, whereas our method makes predictions and adapts the model simultaneously.

**Contrastive learning.** Recently, there has been a surge of interest in contrastive learning Chen et al. (2020); Khosla et al. (2020); Wu et al. (2018); He et al. (2020); Oord et al. (2018), which aims to learn effective representations by aligning positive pairs and separating negative pairs in the embedding space. Due to its ability to learn high-quality representations, contrastive learning has been extensively explored in various fields such as semi-supervised learning Chen et al. (2020); He et al. (2020); Wang et al. (2021b); Zhong et al. (2021), class-imbalanced classification Yang & Xu (2020), TTT Liu et al. (2021b); Chen et al. (2022); Döbler et al. (2022), and domain adaptation Kang et al. (2019); Thota & Leontidis (2021); Chen et al. (2022). In this paper, we analyze the limitation of applying traditional CL in the task of TTT for segmentation. Then, we introduce the solution by adapting the CL to the output space, adjusting the temperature, and simplifying the formulation.

## 3  METHOD

In Section 3.1, we provide a concise introduction to the contrastive loss (CL) and adapt it to the output space, resulting in the Output Contrastive Loss (OCL). Section 3.2 elaborates on the specific implementation of OCL. Moving to Section 3.3, we present the integrated framework, unifying OCL with two established techniques, BN statistics modulation Schneider et al. (2020), and stochastic restoration Wang et al. (2022), aimed at stabilizing the test-time training (TTT) process.

### 3.1  ADAPTING CONTRASTIVE LOSS TO OUTPUT SPACE

The CL has become a widely used technique for representation Chen et al. (2020) and supervised Khosla et al. (2020) learning tasks, as it facilitates the learning of representations by aligning positive

pairs together and separating negative pairs in the embedding space. Given a collection of points $\mathcal{C}$, the positive set $\mathcal{P}_i$, for a given anchor point $i \in \mathcal{C}$, consists of points whose predictions are similar to those of $i$. The CL can then be defined as follows:

$$L = -\mathbb{E}_{i\in\mathcal{C},j\in\mathcal{P}_i} \log \frac{\exp(\text{sim}(\boldsymbol{z}_i, \boldsymbol{z}_j)/\tau)}{\sum_{k\in\mathcal{C}} \mathbb{I}_{[k\neq i]} \exp(\text{sim}(\boldsymbol{z}_i, \boldsymbol{z}_k)/\tau)}, \tag{1}$$

where $\mathbb{I}_{[k\neq i]} \in \{0, 1\}$ is an indicator function evaluating to 1 iff $k \neq i$, $\text{sim}(\boldsymbol{z}_i, \boldsymbol{z}_j)$ denotes the similarity between representation $\boldsymbol{z}_i$ and $\boldsymbol{z}_j$, and $\tau$ is a scalar temperature parameter.

Here, we examine the issue associated with the direct application of the traditional CL formulation to our task and propose solutions. We argue that using the CL to tune the model in the embedding space is sub-optimal. This practice primarily modifies the encoder layers, leaving the classifier head layers unchanged. As a result, merging the modified encoder with the unaltered classifier head can yield unpredictable outcomes. One common solution to this issue involves fine-tuning with labeled data following pre-training with the CL Chen et al. (2020); He et al. (2020). Another solution is to introduce the extra complex regularization designed to dynamically align the task header with the learned representations Chen et al. (2022). However, in the context of TTT for segmentation, fine-tuning with labeled data is unfeasible, and introducing additional complex regularization requires intricate adjustments, limiting the method's versatility. To avoid fine-tuning with labeled data or the complex extra regularization, we propose applying the CL to the output space instead.

In the traditional CL, a small temperature value (e.g., the default $\tau = 0.1$ in SimCLR Chen et al. (2022)) is commonly employed to ensure the uniformity of feature distributions Wang & Liu (2021); Wang & Isola (2020). However, this practice can be detrimental when applying CL to the output space. A low temperature results in a uniform class distribution, which can disrupt the imbalanced class distribution often found in segmentation datasets, as visually demonstrated in Fig. 1. To address this issue, we propose employing a high temperature, which helps maintain the original class distribution. Additionally, employing a high temperature can simplify the CL formulation as follows:

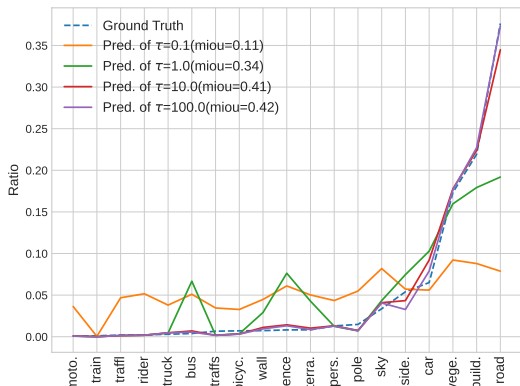

Figure 1: Class ratio when setting different $\tau$ on GTA→Cityscapes.

$$\begin{aligned} L &= \lim_{\tau\to+\infty} \mathbb{E}_{i\in\mathcal{C},j\in\mathcal{P}_i} - \log \frac{\exp(\text{sim}(\boldsymbol{p}_i, \boldsymbol{p}_j)/\tau)}{\sum_{k\in\mathcal{C}} \mathbb{I}_{[k\neq i]} \exp(\text{sim}(\boldsymbol{p}_i, \boldsymbol{p}_k)/\tau)} \\ &= \lim_{\tau\to+\infty} \mathbb{E}_{i\in\mathcal{C},j\in\mathcal{P}_i} - \frac{1}{\tau}\text{sim}(\boldsymbol{p}_i, \boldsymbol{p}_j) + \mathbb{E}_{i,k\in\mathcal{C}}\mathbb{I}_{[k\neq i]}\frac{1}{\tau}\text{sim}(\boldsymbol{p}_i, \boldsymbol{p}_k) + \log(N-1), \end{aligned} \tag{2}$$

where $N$ represents the total number of points in $\mathcal{C}$. $\boldsymbol{p}_i$ refers to the prediction of point $i$, which replaces the representation $\boldsymbol{z}_i$ in Eq. 1. Derivation details of Eq. 2 are present in Supplementary Materials A. By removing two constant values, $\tau$ and $\log(N-1)$, we can write it as:

$$L = \underbrace{-\mathbb{E}_{i\in\mathcal{C},j\in\mathcal{P}_i}\text{sim}(\boldsymbol{p}_i, \boldsymbol{p}_j)}_{\text{positive term}} + \underbrace{\mathbb{E}_{i,k\in\mathcal{C}}\mathbb{I}_{[k\neq i]}\text{sim}(\boldsymbol{p}_i, \boldsymbol{p}_k)}_{\text{negative term}}, \tag{3}$$

In this simplified form, we use positive and negative terms to achieve alignment and separation, respectively. The positive term maximizes the similarity between positive pairs, while the negative term minimizes the similarity between negative pairs. Since this simplified CL formulation is suitable for the output space, we name it as output contrastive loss (OCL). Next, we apply the OCL to our task by defining the positive and the negative terms.

### 3.2 DETAILED IMPLEMENTATION OF OUTPUT CONTRASTIVE LOSS

The traditional CL is applied to the image classification task. Due to the absence of labels, the traditional CL defines the positive set as the different augmented views of the anchor image and the

negative set as different images. This paper studies the segmentation task that necessitates a slight adjustment in how we define the positive and negative sets. We define the positive set as the anchor pixel from different augmented views and define the negative set as the different pixels within the same image. According to this, the positive term in Eq. 3 can be written as:

$$L_{pos} = -\mathbb{E}_i \text{sim}(\boldsymbol{p}_{s1}^i, \boldsymbol{p}_{s2}^i), \tag{4}$$

where $\boldsymbol{p}_{s1}^i$ and $\boldsymbol{p}_{s2}^i$ are prediction probability of $i$-th pixel in two augmented views.

**Augmentations.** Only the horizontal flipping is used to augment the test sample. While stronger augmentations have the potential to enhance a model's generalization to unseen data Xie et al. (2020); Berthelot et al. (2019), they also pose the risk of destroy the knowledge acquired during training. Horizontal flipping, on the other hand, is regarded as a conservative test augmentation technique. Many studies Zou et al. (2018; 2019) have demonstrated its effectiveness in boosting test performance by aggregating predictions from both original and horizontally flipped images. Thus, we chose to focus on horizontal flipping in this work, leaving the exploration of more elaborate augmentation techniques to future research.

The positive term is similar to the well-known consistency loss in aligning the predictions of augmentation views Tarvainen & Valpola (2017); Laine & Aila (2016). However, Eq. 3 suggests that solely relying on this term is not enough to achieve optimal predictions.

Then, the negative term in Eq. 3 is defined as:

$$L_{neg} = \frac{1}{2}\mathbb{E}_{i \neq j}(\text{sim}(\boldsymbol{p}_{s1}^i, \boldsymbol{p}_{s1}^j) + \text{sim}(\boldsymbol{p}_{s2}^i, \boldsymbol{p}_{s2}^j)). \tag{5}$$

Intuitively, reducing the prediction similarity between different pixels can disperse the predictions and prevent the mode collapse Yang et al. (2022), playing a similar role to the diversity loss in the existing studies Chen et al. (2022); Choi et al. (2022); Wang et al. (2021a). This means that the OCL naturally combines the consistency loss and diversity loss.

Because of the high resolution of the output, calculating the similarity for a huge number of pixel pairs in Eq. 5 consumes significant computational and memory resources Zhong et al. (2021). To improve efficiency, we calculate Eq. 5 on a subset of all pixels. For example, we downsample the original output resolution of $1024 \times 512$ by a factor of 8 to a map with a resolution of $128 \times 64$ before performing the calculation Eq. 5.

### 3.3 INTEGRATED FRAMEWORK

Integrating two regularization terms, the overall loss function can be written as:

$$L = \lambda_{pos}L_{pos} + \lambda_{neg}L_{neg}, \tag{6}$$

where $\lambda_{pos}$ and $\lambda_{neg}$ are two loss weights to balance their importance. In practice, we keep the loss weight $\lambda_{neg}$ fixed at 1 and focus on adjusting the $\lambda_{pos}$ to simplify the hyperparameter selection process. The cosine distance is chosen to measure the similarity between two predictions: $\text{sim}(\boldsymbol{p}_i, \boldsymbol{p}_j) = \frac{\boldsymbol{p}_i \cdot \boldsymbol{p}_j}{\|\boldsymbol{p}_i\|_2 \cdot \|\boldsymbol{p}_j\|_2}$. To further improve the adaptation performance, our integrated framework combines the OCL with two near-computation-free techniques, BN Statistics Modulation and Stochastic Restoration.

**BN Statistics Modulation.** BN layer Ioffe & Szegedy (2015) is a fundamental unit in modern DNNs, significantly accelerating convergence while improving the ultimate performance. BN statistics estimated during training time will be used for inference by default. However, these estimated statistics may not accurately reflect the test distribution when applied to out-of-distribution (OOD) data. To address this mismatch, previous studies Li et al. (2016); Schneider et al. (2020) have successfully improved the model's generalization ability by adapting BN statistics to OOD data. Here, we adopt an approach from recent research Schneider et al. (2020) and estimate channel-wise mean and variance $[\boldsymbol{\mu}, \boldsymbol{\sigma}^2]$ by mixing training and test normalization statistics. Specifically, for $\boldsymbol{v} \in \boldsymbol{\mu}, \boldsymbol{\sigma}^2$, we define:

$$\boldsymbol{v} \triangleq \alpha\boldsymbol{v}_{\text{train}} + (1-\alpha)\boldsymbol{v}_{\text{test}}, \tag{7}$$

where $\boldsymbol{v}_{\text{train}}$ and $\boldsymbol{v}_{\text{test}}$ represent the normalization statistics estimated during the training and on currently available test points, respectively. The prior strength $\alpha \in [0, 1]$ controls the trade-off

Table 1: Experimental setting for three benchmarks.

| Benchmark | Task | pre-trained method | Architecture | BN Modu. |
|---|---|---|---|---|
| Synthia→CS | | Source | SegNet-VGG16 | No |
| GTA→CS | TTT | Source, FDA | DeepLab-ResNet101 | Yes |
| | | DAFormer | SegFormer-B5 | No |
| CS→ACDC | continual TTT | Source | SegFormer-B5 | No |

between source and estimated target statistics. It's important to note that the modulation of BN statistics is build on the BN layer. This technique cannot be applied to architectures such as VGG and ViT that lack a BN layer.

**Stochastic Restoration.** To mitigate the error accumulation, we introduce the stochastic restoration that restores the knowledge from the pre-trained model Wang et al. (2022). Specifically, following a gradient update at iteration $t$, the stochastic restoration involves an additional update to the weights $\boldsymbol{W}$ according to the following scheme:

$$\boldsymbol{M} \sim \text{Bernoulli}(p), \tag{8}$$
$$\boldsymbol{W}_{t+1} = \boldsymbol{M} \odot \boldsymbol{W}_0 + (1 - \boldsymbol{M}) \odot \boldsymbol{W}_{t+1}, \tag{9}$$

where $\odot$ represents element-wise multiplication. Here, $p$ is a small probability for restoration, and $\boldsymbol{M}$ is a mask tensor with the same shape as $\boldsymbol{W}_{t+1}$. This mask tensor determines the elements in $\boldsymbol{W}_{t+1}$ that are reverted to the source weight $\boldsymbol{W}_0$.

**Evaluation.** In the OCL framework, both evaluation and adaptation take place concurrently in each iteration. Specifically, when a new test image is presented, the model produces a prediction for that image and subsequently is updated using Eq. 6. It is important to note that the updating process with respect to the current point aims to enhance the model's generalization performance on future data and does not influence the prediction for the current point.

## 4 EXPERIMENTS

### 4.1 EXPERIMENTAL SETUP

**Datasets.** We assess the performance of our framework using two synthetic-to-real benchmarks: GTA Richter et al. (2016) → Cityscapes (CS) Cordts et al. (2016) and Synthia Ros et al. (2016) → CS, as well as one clear-to-adverse-weather benchmark: CS → ACDC Sakaridis et al. (2021). These benchmarks encompass diverse task settings, architectures, and pre-trained methods. You can find the specific details in Table 1.

**Hyperparamters.** In pursuit of a universal hyperparameter configuration adaptable across various scenarios, we employ a consistent set of hyperparameters for all experiments unless otherwise specified. Specifically, we assign a positive loss weight of 3.0, configure the BN modulation prior to 0.85 when the model includes the BN layer (as viewed in Table 1), and set the masking proportion $p$ to 0.01.

### 4.2 MAIN RESULTS

**Results on synthetic-to-real benchmarks.** We evaluate the performance of our method against three simple and commonly used baselines in TTT studies, namely TENT Wang et al. (2020), MEMO Zhang et al. (2021a), and CoTTA Wang et al. (2022). The results of these methods are reproduced using the official code they provide under the same settings. In addition, we provide a comparative analysis of our OCL approach against several DA methods.

In Table 2, Row "Source only" reports the performance of source-pretrained models, and Row "Source only + OCL" reports the results of our OCL. The results demonstrate that our proposed OCL significantly improves model generalization compared to the "Source only" baseline across both benchmarks. Specifically, on the GTA→CS benchmark, our OCL method results in an impressive increase of 7.5 mIoU. On the Synthia→CS benchmark, the mIoU is boosted from 31.5 mIoU to 36.9 mIoU, showcasing a substantial gain of 5.4 mIoU. For the other TTT methods, the MEMO degrades the mIoU of most classes close to zero, indicating mode collapse. The TENT degrades performance slightly compared with the baseline on both two benchmarks, 1.8 mIoU on the GTA→CS

Table 2: Comparison of the mIoU (%) on the Cityscapes validation set for GTA→Cityscapes and Synthia→Cityscapes. All results are based on DeepLab with ResNet101. The gray line reports the results of our OCL.

| Method | Setting | Road | Sidewalk | Building | Wall | Fence | Pole | Light | Sign | Veget. | Terrain | Sky | Person | Rider | Car | Truck | Bus | Train | Motor. | Bicycle | mIoU |
|---|---|---|---|---|---|---|---|---|---|---|---|---|---|---|---|---|---|---|---|---|---|
| **GTA→CS (Val.)** | | | | | | | | | | | | | | | | | | | | | |
| AdaptSegNet | | 86.5 | 36.0 | 79.9 | 23.4 | 23.3 | 23.9 | 35.2 | 14.8 | 83.4 | 33.3 | 75.6 | 58.5 | 27.6 | 73.7 | 32.5 | 35.4 | 3.9 | 30.1 | 28.1 | 42.4 |
| CLAN | | 87.0 | 27.1 | 79.6 | 27.3 | 23.3 | 28.3 | 35.5 | 24.2 | 83.6 | 27.4 | 74.2 | 58.6 | 28.0 | 76.2 | 33.1 | 36.7 | 6.7 | 31.9 | 31.4 | 43.2 |
| AdvEnt | DA | 89.4 | 33.1 | 81.0 | 26.6 | 26.8 | 27.2 | 33.5 | 24.7 | 83.9 | 36.7 | 78.8 | 58.7 | 30.5 | 84.8 | 38.5 | 44.5 | 1.7 | 31.6 | 32.4 | 45.5 |
| CBST | | 91.8 | 53.5 | 80.5 | 32.7 | 21.0 | 34.0 | 28.9 | 20.4 | 83.9 | 34.2 | 80.9 | 53.1 | 24.0 | 82.7 | 30.3 | 35.9 | 16.0 | 25.9 | 42.8 | 45.9 |
| DACS | | 89.9 | 39.7 | 87.9 | 30.7 | 39.5 | 38.5 | 46.4 | 52.8 | 88.0 | 44.0 | 88.8 | 67.2 | 35.8 | 84.5 | 45.7 | 50.2 | 0.0 | 27.3 | 34.0 | 52.1 |
| Source only | - | 71.9 | 15.6 | 74.4 | 22.4 | 14.8 | 22.9 | 35.4 | 18.4 | 81.1 | 22.0 | 68.3 | 57.3 | 27.9 | 68.1 | 33.1 | 5.8 | 6.5 | 30.5 | 35.3 | 37.5 |
| +TENT | | 71.5 | 22.6 | 76.9 | 20.0 | 17.1 | 21.6 | 29.2 | 15.3 | 78.4 | 33.9 | 75.3 | 50.8 | 3.5 | 80.9 | 29.5 | 31.7 | 4.3 | 13.7 | 2.1 | 35.7 |
| +MEMO | TTT | 82.6 | 0.1 | 68.0 | 0.0 | 0.2 | 1.3 | 1.7 | 0.2 | 78.3 | 0.3 | 82.3 | 1.3 | 0.3 | 77.9 | 6.2 | 1.8 | 0.0 | 0.8 | 0.1 | 21.3 |
| +CoTTA | | 74.4 | 13.5 | 75.3 | 24.1 | 14.0 | 22.9 | 31.1 | 16.1 | 81.8 | 22.6 | 69.7 | 57.3 | 26.7 | 71.9 | 33.4 | 6.2 | 8.1 | 27.2 | 31.7 | 37.3 |
| +OCL | | 87.1 | 42.1 | 81.6 | 29.7 | 20.2 | 27.5 | 37.8 | 18.3 | 83.8 | 33.8 | 74.7 | 60.5 | 24.8 | 85.3 | 36.3 | 46.7 | 4.4 | 29.6 | 31.7 | 45.0 |
| FDA | DA | 92.1 | 52.3 | 80.7 | 23.6 | 26.4 | 35.5 | 37.7 | 38.6 | 81.2 | 32.4 | 73.2 | 61.2 | 34.0 | 84.0 | 32.2 | 51.2 | 8.0 | 26.8 | 44.1 | 48.2 |
| +OCL | +TTT | 93.2 | 57.0 | 83.5 | 31.5 | 31.5 | 38.6 | 41.3 | 39.4 | 85.0 | 42.6 | 76.8 | 63.1 | 34.2 | 85.5 | 34.2 | 51.5 | 9.0 | 26.6 | 46.1 | 51.1 |
| DAFormer | DA | 96.5 | 74.0 | 89.5 | 53.4 | 47.7 | 50.6 | 54.7 | 63.6 | 90.0 | 44.4 | 92.6 | 71.8 | 44.8 | 92.6 | 77.8 | 80.6 | 63.6 | 56.7 | 63.4 | 68.8 |
| +OCL | +TTT | 96.6 | 74.7 | 89.6 | 53.5 | 48.1 | 51.3 | 55.3 | 64.0 | 90.0 | 44.5 | 92.5 | 72.3 | 45.4 | 92.8 | 78.6 | 81.4 | 66.8 | 59.0 | 64.0 | 69.5 |
| **Synthia→CS (Val.)** | | | | | | | | | | | | | | | | | | | | | |
| AdvEnt | | 85.6 | 42.2 | 79.7 | 8.7 | 0.4 | 25.9 | 5.4 | 8.1 | 80.4 | - | 84.1 | 57.9 | 23.8 | 73.3 | - | 36.4 | - | 14.2 | 33.0 | 41.2 |
| CBST | DA | 68.0 | 29.9 | 76.3 | 10.8 | 1.4 | 33.9 | 22.8 | 29.5 | 77.6 | - | 78.3 | 60.6 | 28.3 | 81.6 | - | 23.5 | - | 18.8 | 39.8 | 42.6 |
| MRKLD | | 67.7 | 32.2 | 73.9 | 10.7 | 1.6 | 37.4 | 22.2 | 31.2 | 80.8 | - | 80.5 | 60.8 | 29.1 | 82.8 | - | 25.0 | - | 19.4 | 45.3 | 43.8 |
| DACS | | 80.6 | 25.1 | 81.9 | 21.5 | 2.9 | 37.2 | 33.7 | 24.0 | 83.7 | - | 90.8 | 67.6 | 38.3 | 82.9 | - | 38.9 | - | 28.5 | 47.6 | 48.3 |
| Source only | - | 45.2 | 19.6 | 72.0 | 6.7 | 0.1 | 25.4 | 5.5 | 7.8 | 75.3 | - | 81.9 | 57.3 | 17.3 | 39.0 | - | 19.5 | - | 7.0 | 25.7 | 31.5 |
| +TENT | | 38.1 | 18.9 | 57.5 | 1.1 | 0.2 | 24.7 | 7.1 | 9.0 | 74.5 | - | 81.4 | 47.0 | 17.0 | 67.7 | - | 8.6 | - | 5.9 | 29.7 | 30.5 |
| +MEMO | TTT | 63.9 | 0.7 | 65.4 | 0.0 | 0.0 | 2.1 | 0.3 | 0.3 | 66.4 | - | 78.1 | 6.7 | 0.5 | 15.5 | - | 0.8 | - | 0.5 | 0.1 | 19.0 |
| +CoTTA | | 48.5 | 20.8 | 73.1 | 8.4 | 0.2 | 24.3 | 12.6 | 11.0 | 76.0 | - | 82.2 | 56.6 | 17.3 | 40.2 | - | 21.1 | - | 9.2 | 27.7 | 33.0 |
| +OCL | | 66.6 | 27.5 | 78.8 | 8.0 | 0.2 | 29.0 | 8.1 | 11.3 | 80.1 | - | 82.4 | 55.9 | 16.5 | 58.9 | - | 28.3 | - | 11.8 | 28.4 | 36.9 |
| FDA | DA | 76.2 | 33.3 | 74.8 | 8.3 | 0.3 | 32.2 | 19.8 | 24.5 | 62.6 | - | 83.8 | 58.2 | 27.3 | 82.2 | - | 40.3 | - | 31.5 | 45.1 | 43.8 |
| +OCL | +TTT | 78.0 | 33.8 | 78.9 | 10.9 | 0.3 | 34.1 | 21.9 | 26.1 | 75.7 | - | 84.8 | 60.8 | 28.6 | 84.3 | - | 43.1 | - | 32.5 | 45.3 | 46.2 |
| DAFormer | DA | 82.2 | 37.2 | 88.6 | 42.9 | 8.5 | 50.1 | 55.1 | 54.3 | 85.7 | - | 88.0 | 73.6 | 48.6 | 87.6 | - | 62.8 | - | 53.1 | 62.4 | 61.3 |
| +OCL | +TTT | 81.6 | 36.5 | 88.7 | 43.1 | 8.4 | 50.8 | 55.8 | 55.1 | 86.2 | - | 88.4 | 74.2 | 49.5 | 87.8 | - | 63.2 | - | 54.5 | 62.8 | 61.7 |

and 1.0 mIoU on the Synthia→CS. The CoTTA achieves comparable performance to the baseline, improving 1.5 mIoU on the Synthia→CS but decreasing 0.2 mIoU on the GTA→CS.

We also compare our proposed OCL method with several existing DA methods. Some classical DA methods are selected as our baselines, including AdaptSegNet Tsai et al. (2018), CLAN Luo et al. (2019), AdvEnt Vu et al. (2019), CBST Zou et al. (2018), MRKLD Zou et al. (2019), and DACS Tranheden et al. (2021). It's important to note that the TTT setting is inherently more challenging than standard DA due to its real-time decision-making and online learning requirements. In the GTA→CS benchmark, our proposed OCL method even outperforms some methods that are specifically designed for DA segmentation tasks, such as AdaptSegNet and CLAN. However, on the Synthia→CS benchmark, our OCL's performance is understandably not as competitive as dedicated DA methods.

**Results on the clear-to-adverse-weather benchmark.** In the CS→ACDC benchmark, we establish a continual TTT scenario following Wang et al. (2022). This involves repeating the same sequence group, comprising the four weather conditions (Fog→Night→Rain→Snow), a total of 10 rounds (resulting in a sequence like Fog→Night→Rain→Snow→Fog...). The adaptation results obtained during the first, fourth, seventh, and last rounds are summarized in Table 3. These results underscore the stable improvement in model performance facilitated by our method in this continuous setting. When considering all rounds and weather conditions, our proposed method achieves 58.9 mIoU, surpassing the baseline by 2.2 mIoU. Then, the average performance in the tenth round remains similar to that in the first round. In contrast, the TENT method exhibits a continuous degradation in performance as the rounds progress. Moreover, when compared with CoTTA, our OCL demonstrates better performance in more challenging scenarios like "night" and "snow" while delivering slightly lower performance in easier scenarios like "fog". The overall performance superiority makes our OCL a robust choice for continual TTT in complex and evolving environments.

**Results for combining DA methods and OCL.** As our baseline, we select models pre-trained using FDA Yang & Soatto (2020) and DAFormer Hoyer et al. (2022a). FDA is a straightforward DA technique that addresses domain shift by manipulating the spectral characteristics of the source image. When applying our OCL method to these pre-trained models, we observed notable improvements in mIoU. Specifically, as shown in Table 2, our OCL improved mIoU by 2.9 percent on GTA→CS

Table 3: Comparison of the mIoU (%) on the Cityscapes-to-ACDC online continual test-time adaptation task. We evaluate the four test conditions continually for ten rounds to evaluate the long-term adaptation performance. All results are evaluated based on the Segformer-B5 architecture.

| Time | $t$ | | | | | | | | | | | | | | | $\rightarrow$ |
|---|---|---|---|---|---|---|---|---|---|---|---|---|---|---|---|---|
| Round | 1 | | | | 4 | | | | 7 | | | | 10 | | | | All |
| Condition | Fog | Night | rain | snow | Fog | Night | rain | snow | Fog | Night | rain | snow | Fog | Night | rain | snow | Mean |
| Source | 69.1 | 40.3 | 59.7 | 57.8 | 69.1 | 40.3 | 59.7 | 57.8 | 69.1 | 40.3 | 59.7 | 57.8 | 69.1 | 40.3 | 59.7 | 57.8 | 56.7 |
| BN Stats Adapt | 62.3 | 38.0 | 54.6 | 53.0 | 62.3 | 38.0 | 54.6 | 53.0 | 62.3 | 38.0 | 54.6 | 53.0 | 62.3 | 38.0 | 54.6 | 53.0 | 52.0 |
| TENT-continual | 69.0 | 40.2 | 60.1 | 57.3 | 66.5 | 36.3 | 58.7 | 54.0 | 64.2 | 32.8 | 55.3 | 50.9 | 61.8 | 29.8 | 51.9 | 47.8 | 52.3 |
| CoTTA | **70.9** | 41.2 | 62.4 | 59.7 | **70.9** | 41.0 | **62.7** | 59.7 | **70.9** | 41.0 | **62.8** | 59.7 | **70.8** | 41.0 | 62.8 | 59.7 | 58.6 |
| OCL (ours) | 70.2 | **42.7** | **62.5** | **61.0** | 69.4 | **42.7** | 62.1 | **60.2** | 69.7 | **42.9** | 62.0 | **60.5** | 69.7 | **42.9** | **62.9** | **60.8** | **58.9** |

and 2.4 percent on Synthia→CS compared to the FDA baseline. On the other hand, DAFormer represents a more advanced DA approach that enhances network architectures and training strategies. Table 2 shows that the performance gains achieved with DAFormer were 0.7 mIoU on GTA→CS and 0.4 mIoU on Synthia→CS, for the same two benchmarks. In summary, our findings indicate that OCL can effectively complement existing DA methods, particularly when the pre-trained model is not well adapted to the test data. Even as the model's generalization ability improves, OCL continues to provide slight performance gains.

**Results on FCN-8s-VGG16.** We replaced the segmentation model with an FCN based on the VGG16 architecture. The results, as shown in Table 4, indicate that our proposed OCL method led to a notable improvement in mIoU. Specifically, we observed an increase of 2.5 percent on the GTA→CS benchmark and 1.5 percent on the Synthia→CS benchmark.

Table 4: Results of GTA→CS and Synthia→CS pre-trained on SegNet-VGG16 network.

| Method | GTA→CS | Synthia→CS |
|---|---|---|
| Source only | 29.9 | 26.5 |
| +OCL | 32.4(+2.5) | 28.0(+1.5) |

### 4.3 ABLATION STUDY

In this section, we perform a comprehensive examination to further assess the individual effectiveness of each component within our proposed method. Additionally, we investigate the model's sensitivity to changes in hyperparameters. These analyses are carried out on the synthetic-to-real benchmarks using the DeepLab-ResNet101 architecture.

**Components Ablation.** We conducted an empirical analysis to assess the impact of our design choices and to dissect the effects of different components within the integrated framework. The results of this analysis are presented in Table 5. By comparing Row 1 and 2, it's evident that relying solely on the OCL effectively enhances model performance, a noteworthy outcome in these challenging scenarios. Rows 3 and 4 further improve performance when building upon the results of Row 2, demonstrating that the OCL is compatible with BN statistic modulation and stochastic restoration. In comparing Row 5 with the preceding four rows, it becomes evident that the combination of all three components yields the best performance.

Table 5: Ablation of the components of the integrated framework on GTA→CS and Synthia→CS. BN. and S.R. denotes BN statistic Modulation and stochastic restoration, respectively.

| No | OCL | BN. | S.R. | GTA→CS | Synthia→CS |
|---|---|---|---|---|---|
| 1 | - | - | - | 37.5 | 31.5 |
| 2 | ✓ | - | - | 39.6(+2.1) | 32.8(+1.3) |
| 3 | ✓ | ✓ | - | 42.1(+4.6) | 35.2(+3.7) |
| 4 | ✓ | - | ✓ | 40.9(+3.4) | 34.1(+2.6) |
| 5 | ✓ | ✓ | ✓ | 45.0(+7.5) | 36.9(+5.4) |

**Sensitivity Analysis of Hyperparameters.** In our sensitivity analysis, we explore how different hyperparameter choices affect the performance of our integrated method. Specifically, we investigate the impact of three hyperparameters: BN prior ($\alpha$), masking proportion ($p$), and positive weight ($\lambda_{pos}$). The results of this analysis are presented in Table 6.

Upon comparing the performances under varying hyperparameters, we observe that our method is most sensitive to changes in $\lambda_{pos}$. When $\lambda_{pos} = 1$, there is a significant drop in performance. This occurs because the negative term in Eq. 6 becomes dominant, distorting the class distribution. Conversely, setting $\lambda_{pos}$ to a high value yields opposing effects on the two benchmarks: enhancing performance on Synthia→CS while degrading it on GTA→CS. Hence, choosing an appropriate $\lambda_{pos}$ is crucial for achieving optimal results on diverse scenarios..

Table 6: Parameter study of the BN prior $\alpha$, Masking proportion $p$, and positive weight $\lambda_{pos}$ on Synthetic-to-Real benchmarks using the DeepLab-ResNet101. The color indicates the performance differences compared to default hyperparameters ($\alpha = 0.85$, $p = 0.01$, $\lambda_{pos} = 3.0$). The performance achieved with the default hyperparameters listed in the last column for reference.

| Benchmark | BN prior $\alpha$ | | | | Masking proportion $p$ | | | | positive weight $\lambda_{pos}$ | | | | Default |
|---|---|---|---|---|---|---|---|---|---|---|---|---|---|
| | 0.75 | 0.8 | 0.9 | 0.95 | 0.005 | 0.015 | 0.02 | 0.03 | 1.0 | 2.0 | 5.0 | 6.0 | |
| GTA→CS | 44.0 | 44.6 | 45.5 | 45.4 | 44.5 | 45.4 | 45.3 | 44.9 | 36.6 | 43.2 | 44.8 | 44.0 | 45.0 |
| Synthia→CS | 37.5 | 37.3 | 36.2 | 35.4 | 36.1 | 37.3 | 37.4 | 37.3 | 35.4 | 36.3 | 38.2 | 38.7 | 36.9 |

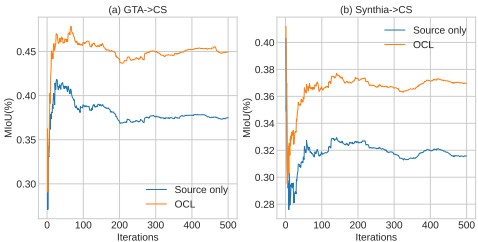

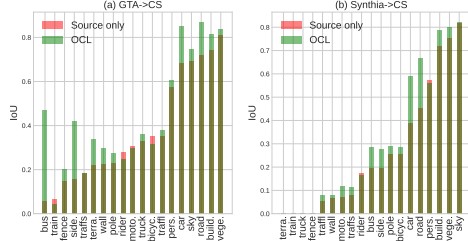

Figure 2: The accumulated MIoUs (i.e., MIoUs is calculated over all previous test samples) during adaptation process.

Figure 3: Class-wise Performance comparison on GTA→CS and Synthia→CS with DeepLab-ResNet101 network.

Regarding $\alpha$, we note an interesting contrast in its impact on the two benchmarks. A larger $\alpha$ leads to improved performance in the GTA→CS scenario, while a smaller $\alpha$ yields better results in the Synthia→CS scenario. This suggests that Synthia→CS experiences a more significant domain shift than GTA→CS.

Finally, we examine the effects of $p$. Both extremely small and extremely large values for $p$ negatively affect performance. This underscores the importance of preserving moderate information from the pre-trained model to maintain good results.

### 4.4 FURTHER ANALYSIS

**MIoUs changing.** To analyze the evolving performance trend throughout the adaptation process, we monitor the trend of the accumulated MIoU during the adaptation. The results are reported in Fig. 2. Our observations reveal that the proposed OCL significantly boosts the model's generalization ability, especially in the early stages of adaptation (within the first 30 iterations). Then, it continues to enhance performance gradually during subsequent adaptation phases across both benchmark datasets. Furthermore, the cumulative mIoU curve of OCL follows a similar trajectory to the cumulative mIoU curve of the pre-trained model, highlighting that OCL retains certain information from the pre-trained model.

**Class-wise performance comparison.** We further discuss the strengths and bottlenecks of our OCL by analyzing the class-wise performance. The results are shown in Fig. 3. We find that the OCL can improve the performance across majority classes. Specifically, the OCL improves the IoU on 15 out of 19 classes in GTA→CS whereas the OCL improves the IoU on 14 out of 16 classes in Synthia→CS. For the classes with degraded performance, we find that them are easily confused in semantic, like 'Rider' and 'Person' classes.

## 5 CONCLUSION

This paper investigates a practical and challenging task of TTT for semantic segmentation, which enable models to quickly adapt to new domains during evaluation. To stabilize the adaptation process, we present a simple and novel loss, OCL, that adapts the CL to our task. Specifically, we extend the CL from the representation space to the output space, utilizing a high temperature and simplifying the formulation. Our comprehensive experimentation consistently verifies the effectiveness of the OCL across multiple evaluation scenarios. These results underline the potential of the OCL as a valuable tool in the quest for more robust and adaptable semantic segmentation models, bridging the gap between training and real-world deployment in diverse environmental conditions.

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

# A  MORE DEVIATION DETAILS OF EQUALITY 2

$$
\begin{aligned}
L &= \lim_{\tau \to +\infty} \mathbb{E}_{i \in \mathcal{C}, j \in \mathcal{P}_i} - \log \frac{\exp(\mathrm{sim}(\boldsymbol{p}_i, \boldsymbol{p}_j)/\tau)}{\sum_{k \in \mathcal{C}} \mathbb{I}_{[k \neq i]} \exp(\mathrm{sim}(\boldsymbol{p}_i, \boldsymbol{p}_k)/\tau)} \\
&= \lim_{\tau \to +\infty} \mathbb{E}_{i \in \mathcal{C}, j \in \mathcal{P}_i} - \frac{1}{\tau} \mathrm{sim}(\boldsymbol{p}_i, \boldsymbol{p}_j) + \mathbb{E}_{i \in \mathcal{C}} \log \sum_{k \in \mathcal{C}} \mathbb{I}_{[k \neq i]} \exp(\mathrm{sim}(\boldsymbol{p}_i, \boldsymbol{p}_k)/\tau) \\
&= \lim_{\tau \to +\infty} \mathbb{E}_{i \in \mathcal{C}, j \in \mathcal{P}_i} - \frac{1}{\tau} \mathrm{sim}(\boldsymbol{p}_i, \boldsymbol{p}_j) + \mathbb{E}_{i \in \mathcal{C}} \log \left( \frac{\sum_{k \in \mathcal{C}} \mathbb{I}_{[k \neq i]} \exp(\mathrm{sim}(\boldsymbol{p}_i, \boldsymbol{p}_k)/\tau)}{N-1} * (N-1) \right) \\
&= \lim_{\tau \to +\infty} \mathbb{E}_{i \in \mathcal{C}, j \in \mathcal{P}_i} - \frac{1}{\tau} \mathrm{sim}(\boldsymbol{p}_i, \boldsymbol{p}_j) + \log(N-1) + \mathbb{E}_{i \in \mathcal{C}} \frac{\sum_{k \in \mathcal{C}} \mathbb{I}_{[k \neq i]} \exp(\mathrm{sim}(\boldsymbol{p}_i, \boldsymbol{p}_k)/\tau)}{N-1} - 1 \\
&= \lim_{\tau \to +\infty} \mathbb{E}_{i \in \mathcal{C}, j \in \mathcal{P}_i} - \frac{1}{\tau} \mathrm{sim}(\boldsymbol{p}_i, \boldsymbol{p}_j) + \log(N-1) + \mathbb{E}_{i \in \mathcal{C}} \frac{\sum_{k \in \mathcal{C}} \mathbb{I}_{[k \neq i]} \mathrm{sim}((1 + \boldsymbol{p}_i, \boldsymbol{p}_k)/\tau)}{N-1} - 1 \\
&= \lim_{\tau \to +\infty} \mathbb{E}_{i \in \mathcal{C}, j \in \mathcal{P}_i} - \frac{1}{\tau} \mathrm{sim}(\boldsymbol{p}_i, \boldsymbol{p}_j) + \mathbb{E}_{i, k \in \mathcal{C}} \mathbb{I}_{[k \neq i]} \frac{1}{\tau} \mathrm{sim}(\boldsymbol{p}_i, \boldsymbol{p}_k) + \log(N-1),
\end{aligned}
\tag{10}
$$

where the fourth step and fifth step utilize the tylor expansion of the $\log()$ and $\exp()$, respectively.

# B  MORE EXPERIMENTAL DETAILS

## B.1  DATASETS

Our evaluation encompasses a total of four datasets, each of which is introduced below:

**Cityscapes.** Cityscapes Cordts et al. (2016) is a dataset consisting of 5,000 densely annotated images with a resolution of $2048 \times 1024$. In the GTA5→CS and Synthia→CS benchmarks, we utilize 500 validation images as the unseen target domain. Following the standard domain adaptation (DA) setting Yang & Soatto (2020), the images are resized to $1024 \times 512$, with no random cropping.

**GTA5.** GTA5 Richter et al. (2016) is a dataset comprising 24,966 synthetic images extracted from a video game, each with a resolution of $1914 \times 1052$. It shares 19 classes with Cityscapes. In line with the standard DA setting Hoyer et al. (2022a), we resize the images to $1280 \times 720$.

**Synthia.** We employ the SYNTHIA-RANDCITYSCAPES partition from the Synthia dataset Ros et al. (2016), which includes 9,400 images with a resolution of $1280 \times 760$. These images share 16 classes with Cityscapes.

**ACDC.** ACDC Sakaridis et al. (2021) is a real-world street scene dataset with 19 common categories overlapping with Cityscapes. It includes images captured under adverse conditions such as foggy, nighttime, rainy, and snowy scenarios. For adaptation, we use 400 unlabeled images from each adverse condition.

Table 7: The origin of pre-trained model checkpoints.

| Benchmark | Pre-trained method | Architecture | Provider | Url |
|---|---|---|---|---|
| GTA→CS | Source | SegNet-VGG16 | FDA Yang & Soatto (2020) | https://drive.google.com/open?id=1pgHtwBKUcbAyItnU4hgMb96UfY1PGiCv&authuser=0 |
| | Source | DeepLab-ResNet101 | MaxSquareChen et al. (2019) | https://drive.google.com/open?id=1KP37cQo_9NEBczm7pvq_zEmmosdhxvlF&authuser=0 |
| | FDA | DeepLab-ResNet101 | FDAYang & Soatto (2020) | https://drive.google.com/open?id=1HueawBlg6RFaKNt2wAX__1vmmupKqHmS&authuser=0 |
| | DAFormer | SegFormer-B5 | DAFormerHoyer et al. (2022a) | https://drive.google.com/open?id=1pG3kDClZDGwp1vSTEXmTchkGHmnLQNdP&authuser=0 |
| Synthia→CS | Source | SegNet-VGG16 | FDAYang & Soatto (2020) | https://drive.google.com/open?id=1KP37cQo_9NEBczm7pvq_zEmmosdhxvlF&authuser=0 |
| | Source | DeepLab-ResNet101 | MaxSquareChen et al. (2019) | https://drive.google.com/open?id=1wLffQRljXK1xoqRY64INvb2lk2ur5fEL&authuser=0 |
| | FDA | DeepLab-ResNet101 | FDAYang & Soatto (2020) | https://drive.google.com/open?id=1FRI_KIWnubyknChhTOAVl6ZsPxzvEXce&authuser=0 |
| | DAFormer | SegFormer-B5 | DAFormerHoyer et al. (2022a) | https://drive.google.com/open?id=1V9EpoTePjGq33B8MfombxEEcq9a2rBEt&authuser=0 |
| CS→ACDC | Source | SegFormer-B5 | CoTTAWang et al. (2022) | https://drive.qin.ee/api/raw/?path=/cv/cvpr2022/acdc-seg.tar.gz |

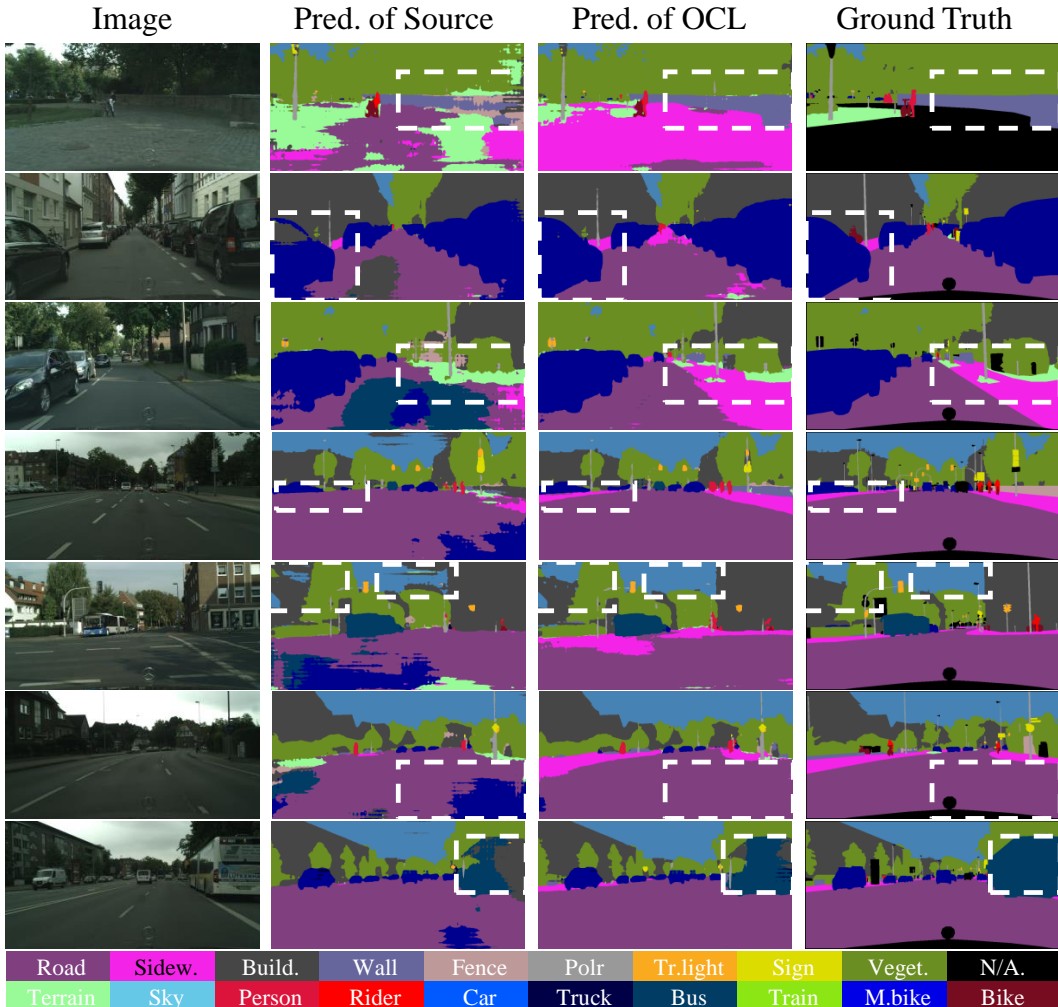

| Image | Pred. of Source | Pred. of OCL | Ground Truth |

| Road | Sidew. | Build. | Wall | Fence | Polr | Tr.light | Sign | Veget. | N/A. |
| Terrain | Sky | Person | Rider | Car | Truck | Bus | Train | M.bike | Bike |

Figure 4: Illustrative predictions, organized into seven rows, showcasing enhanced segmentation accuracy for wall, car, terrain, sidewalk, sky, road, and bus. These results were obtained through experimentation on the GTA→CS benchmark. For a closer look, refer to the white dotted box.

## B.2 IMPLEMENTATION DETAILS

The Test-Time Training (TTT) process is composed of two main phases: pre-training and fine-tuning. Let's delve into each of these phases.

**Pre-training.** The outcome of TTT is strongly influenced by the performance of the pre-trained model. To ensure consistency with prior studies, we opt to utilize pre-trained model checkpoints provided in their respective research works. Table 7 provides information regarding the origins of all pre-trained models utilized in our study.

**Fine-tuning.** The proposed Output Contrastive Loss (OCL) necessitates an optimization process. For the GTA→CS and Synthia→CS benchmarks, we employ the SGD optimizer with a learning rate of 2e-5 for GTA→Cityscapes and 1e-5 for Synthia→Cityscapes, a momentum value of 0.9, and a weight decay rate of 5e-4. In the case of the CS→ACDC benchmark, following the CoTTA Wang et al. (2022) setting, we use the Adam optimizer with a learning rate of 0.0006/8, $\beta_1$ of 0.9, and $\beta_2$ of 0.999.

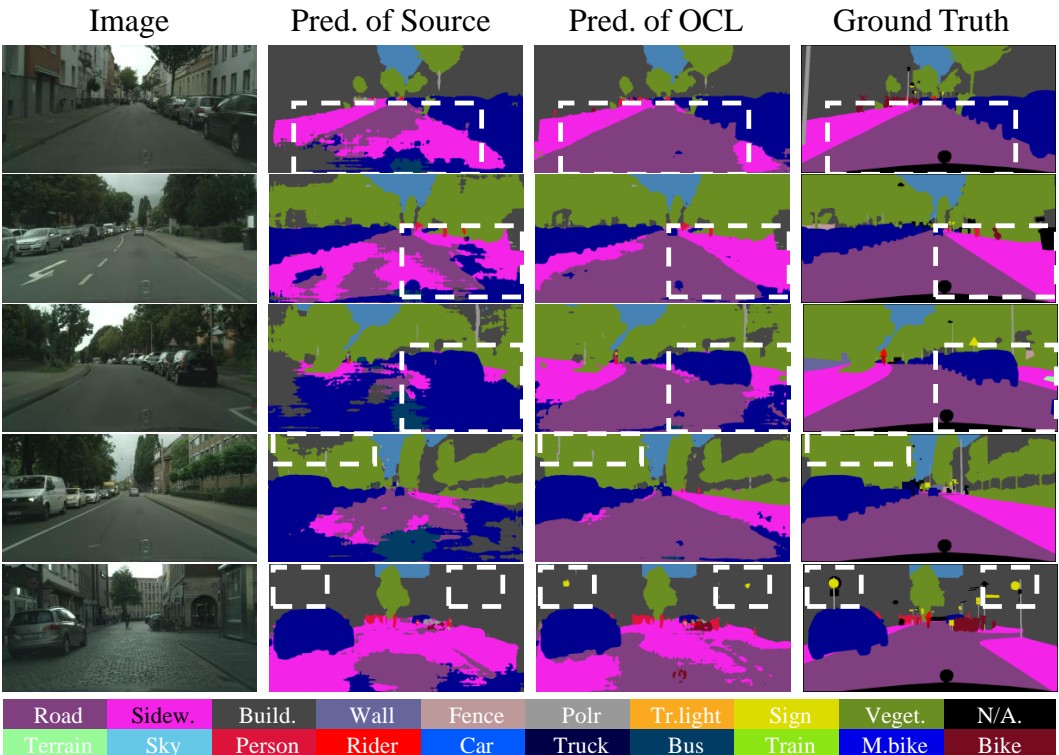

Figure 5: Illustrative predictions, organized into five rows, showcasing enhanced segmentation accuracy for road, sidewalk, cars, trees, and sign. These results were obtained through experimentation on the Synthia→CS benchmark. For a closer look, refer to the white dotted box.

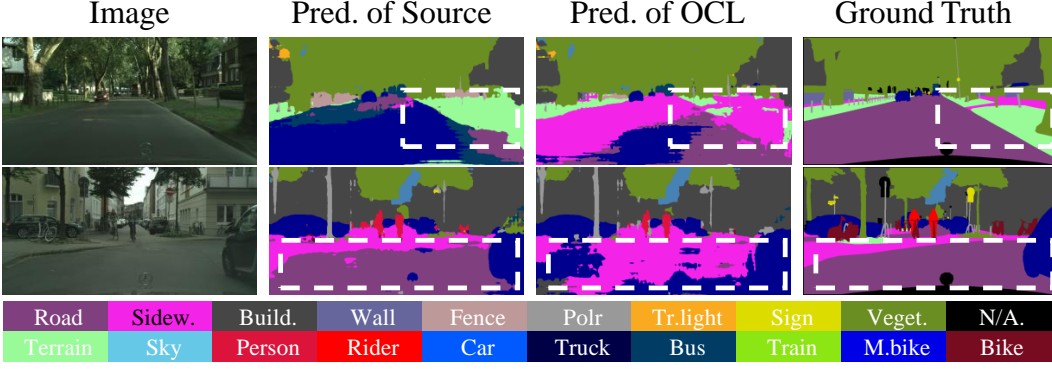

Figure 6: Failure cases of OCL on GTA→CS and Synthia→CS benchmarks, presented in two rows. In these cases, the OCL tends to misclassify other classes as sidewalks.

## C  MORE EXPERIMENTAL RESULTS

### C.1  QUALITATIVE RESULTS

Supplementing the example predictions in the main paper, we show further representative examples of the strength and weaknesses of OCL in comparison with the baseline. Next, first two paragraphs introduce examples with improved performance on the GTA→CS and Synthia→CS, respectively. The third paragraph introduces the failure cases on two benchmarks.

On the GTA→CS benchmark, we present seven examples demonstrating improved segmentation for the following classes: wall, car, terrain, sidewalk, sky, road, and bus (Fig. 4). Here's a breakdown

of these improvements: Wall (Fig. 4, first row): OCL recognizes entire walls while the pre-trained model misclassifies parts as fences. Car (Fig. 4, second row): The pre-trained model fails to recognize car windows, but OCL corrects this. Terrain (Fig. 4, third row): The pre-trained model confuses terrain and sidewalk, while OCL correctly distinguishes between them. Sidewalk (Fig. 4, fourth row): The pre-trained model mixes up sidewalks and roads, but OCL correctly identifies them. Sky (Fig. 4, fifth row): The pre-trained model confuses sky and buildings, but OCL accurately distinguishes between them. Road (Fig. 4, sixth row): The pre-trained model falsely predicts roads as cars, whereas OCL correctly classifies them. Bus (Fig. 4, seventh row): The pre-trained model misclassifies parts of buses as buildings, while OCL correctly classifies them.

On the Synthia→CS benchmark, we provide five examples highlighting improved segmentation for the following classes: road, sidewalk, cars, trees, and signs (Fig. 5). Here's a detailed breakdown of these improvements: Road (Fig. 4, first row): The pre-trained model incorrectly classifies most roads as sidewalks and cars, while OCL corrects these misclassifications. Sidewalk (Fig. 4, second row): The pre-trained model fails to recognize entire sidewalks, but OCL successfully identifies them. Car (Fig. 4, third row): The pre-trained model misclassifies other classes as cars, but OCL rectifies these errors. Tree (Fig. 4, fourth row): The pre-trained model confuses trees with buildings, but OCL correctly distinguishes between them. Sign (Fig. 4, fifth row): The pre-trained model struggles to recognize signs within buildings, while OCL successfully identifies a portion of them.

Fig. 6 shows one common failure mode seen in the OCL, that is the OCL may misclassify the classes like terrain (first row) and road (second row) as the sidewalk.

## C.2 Experiment Results on CS→ACDC

The full experimental results for the continual test-time adaptation task from CS→ACDC are provided in Table 8. These experiments clearly demonstrate that the proposed OCL model consistently maintains strong performance over extended periods.

Table 8: Semantic segmentation results (mIoU in %) on the CS-to-ACDC online continual test-time adaptation task. We evaluate the four test conditions continually for ten times to evaluate the long-term adaptation performance. All results are evaluated based on the Segformer-B5 architecture.

| Time | $t$ | | | | | | | | | | | | | | | | | | | → |
|---|---|---|---|---|---|---|---|---|---|---|---|---|---|---|---|---|---|---|---|---|
| Condition | Fog | Night | rain | snow | Fog | Night | rain | snow | Fog | Night | rain | snow | Fog | Night | rain | snow | Fog | Night | rain | snow | cont. |
| Round | 1 | | | | 2 | | | | 3 | | | | 4 | | | | 5 | | | | cont. |
| Source | 69.1 | 40.3 | 59.7 | 57.8 | 69.1 | 40.3 | 59.7 | 57.8 | 69.1 | 40.3 | 59.7 | 57.8 | 69.1 | 40.3 | 59.7 | 57.8 | 69.1 | 40.3 | 59.7 | 57.8 | cont. |
| BN Stats Adapt | 62.3 | 38.0 | 54.6 | 53.0 | 62.3 | 38.0 | 54.6 | 53.0 | 62.3 | 38.0 | 54.6 | 53.0 | 62.3 | 38.0 | 54.6 | 53.0 | 62.3 | 38.0 | 54.6 | 53.0 | cont. |
| Tent-continual | 69.0 | 40.2 | 60.1 | 57.3 | 68.3 | 39.0 | 60.1 | 56.3 | 67.5 | 37.8 | 59.6 | 55.0 | 66.5 | 36.3 | 58.7 | 54.0 | 65.7 | 35.1 | 57.7 | 53.0 | cont. |
| COTTA | 70.9 | 41.2 | 62.4 | 59.7 | 70.9 | 41.1 | 62.6 | 59.7 | 70.9 | 41.0 | 62.7 | 59.7 | 70.9 | 41.0 | 62.7 | 59.7 | 70.9 | 41.0 | 62.8 | 59.7 | cont. |
| OCL | 70.2 | 42.7 | 62.5 | 61.0 | 69.6 | 42.7 | 62.6 | 60.9 | 69.4 | 42.6 | 62.8 | 60.6 | 69.4 | 42.7 | 62.1 | 60.2 | 69.3 | 42.6 | 62.9 | 60.6 | cont. |
| Round | 6 | | | | 7 | | | | 8 | | | | 9 | | | | 10 | | | | Mean |
| Source | 69.1 | 40.3 | 59.7 | 57.8 | 69.1 | 40.3 | 59.7 | 57.8 | 69.1 | 40.3 | 59.7 | 57.8 | 69.1 | 40.3 | 59.7 | 57.8 | 69.1 | 40.3 | 59.7 | 57.8 | 56.7 |
| BN Stats | 62.3 | 38.0 | 54.6 | 53.0 | 62.3 | 38.0 | 54.6 | 53.0 | 62.3 | 38.0 | 54.6 | 53.0 | 62.3 | 38.0 | 54.6 | 53.0 | 62.3 | 38.0 | 54.6 | 53.0 | 52.0 |
| Tent-continual | 64.9 | 34.0 | 56.5 | 52.0 | 64.2 | 32.8 | 55.3 | 50.9 | 63.3 | 31.6 | 54.0 | 49.8 | 62.5 | 30.6 | 52.9 | 48.8 | 61.8 | 29.8 | 51.9 | 47.8 | 52.3 |
| CoTTA | 70.9 | 41.0 | 62.8 | 59.7 | 70.9 | 41.0 | 62.8 | 59.7 | 70.9 | 41.0 | 62.8 | 59.7 | 70.8 | 41.0 | 62.8 | 59.7 | 70.8 | 41.0 | 62.8 | 59.7 | 58.6 |
| OCL | 69.3 | 42.6 | 62.7 | 60.9 | 69.7 | 42.9 | 62.0 | 60.5 | 70.2 | 41.6 | 62.3 | 60.7 | 69.4 | 42.7 | 62.8 | 60.8 | 69.7 | 42.9 | 62.9 | 60.8 | 58.9 |

## C.3 Computation Efficiency

In real-time adaptation scenarios, computational efficiency is a crucial consideration. This involves not only the ability to adapt quickly to data streams but also the efficient use of GPU memory, which can be vital for practical applications. Table 9 provides information about GPU memory usage and processing times for various methods. Compared to the conventional testing process, our OCL exhibits a similar GPU memory footprint but significantly increased processing times. The primary reason for this discrepancy is that our OCL entails both forward and backward processes for the input image and its augmented view, while the traditional testing process only requires a forward pass for the input image.

Table 9: Efficiency comparison for processing 500 images with resolution $512 \times 1024$ on DeepLab-ResNet101.

| Method | GPU Memo. | Times |
|---|---|---|
| Baseline | 8.1G | 59secs |
| OCL(our) | 10.1G | 339secs |

# D LIMITATION

One limitation of the OCL lies in its potential inefficiency, which may hinder its application in real-time scenarios. As shown in Table 9, our OCL takes approximately five times longer than the regular testing process. To enhance efficiency, future work could focus on selecting a subset of informative samples for model adaptation.

Furthermore, there is room for investigation regarding the choice of augmentation techniques. Augmentation plays a vital role in enhancing a model's generalization to unseen domains, but in our study, we restricted ourselves to using only basic horizontal flipping as an augmentation technique. This simplistic form of augmentation has its limitations in terms of improving generalization. On the other hand, it's essential to consider task-specific knowledge and how it can be incorporated through the use of more specialized augmentation techniques. For instance, when adapting a model from a daytime domain to a nighttime one, employing color augmentation could substantially enhance the model's ability to generalize effectively.

