# OpenReview forum: "Test-Time Training for Semantic Segmentation with Output Contrastive Loss"
_ICLR.cc/2024/Conference — ICLR 2024 Conference Withdrawn Submission_

### Official Review · Reviewer_Bbfh · 2023-10-29

**Soundness:** 2 fair
**Presentation:** 2 fair
**Contribution:** 2 fair
**Rating:** 3
**Confidence:** 3

**Summary:**

This paper tackles a challenging task: test-time training for semantic segmentation. The authors propose the so-called output constrastive loss. By combining the proposed loss with two existing techniques: BN statistics modulation and stochastic restoration, the proposed method achieves promising results on several datasets. The authors also conduct some analysis and ablation studies to validate some design choices.

**Strengths:**

1. Overall, the proposed method is easy to understand and well-performed
2. The proposed method can be combined with source-only baseline and DA methods, on several different segmentation network architectures

**Weaknesses:**

**1. Method.**
- The authors claim that “using the CL to tune the model in the embedding space is sub-optimal”. However, there is no such experiments to support this claim. What will be the performance if we apply CL in the embedding space instead?
- Also, instead of applying CL in the embedding space (between encoder and decoder), we can also apply CL in the second last layer (also some feature space). The claim “this practice primarily modifies the encoder layers, leaving the classifier head layers unchanged” does not seem correct to me.
- The authors claim that using a high temperature in the output contrastive loss will avoid too uniform class distribution. However, wouldn’t this increase the confirmation bias if the label distributions between source and target are very different? Is it possible that this method can work is due to the label distribution difference between source and target domains is small?
- From Table 5, it seems that most of the performance gains actually come from BN + SR, and neither of them is this paper’s contribution. Would also like to see the numbers of BN + SR only, which is actually the strong baseline we should compare with.
- Similarly, since BN and SR are not the core contributions of this paper, I don’t think the authors should show parameters study for them in the main paper (Table 6).

**2. Presentation.**
- What exactly do the authors want to convey in Figure 1? I think we should have the label distribution of GTA and Cityscapes for comparison here.
- In Section 3.3, the claim “we introduce the stochastic restoration …” is problematic. It sounds like the authors propose this technique. But actually not.
- Again, it is good to show label distribution in Figure 3.

**3. Related work.**
Some important related works are missing. Please properly cite them and update the related work section accordingly.

Contrastive loss and consistency loss for semantic segmentation.
- Wang et al., “Dense Contrastive Learning for Self-Supervised Visual Pre-Training”. CVPR 2021
- Xie et al., “Propagate Yourself: Exploring Pixel-Level Consistency for Unsupervised Visual Representation Learning”. CVPR 2021
- Zhao et al., “Contrastive Learning for Label-Efficient Semantic Segmentation”. ICCV 2021
- Wang et al., “Exploring Cross-Image Pixel Contrast for Semantic Segmentation”. ICCV 2021

BN statistics modulation.
- Nado et al., “Evaluating Prediction-Time Batch Normalization for Robustness under Covariate Shift”
- Khurana et al., “SITA: Single Image Test-time Adaptation”
- Zou et al., “Learning Instance-Specific Adaptation for Cross-Domain Segmentation”. ECCV 2022

**Questions:**

Please see the weakness section.

---

### Official Review · Reviewer_hEgU · 2023-11-01

**Soundness:** 2 fair
**Presentation:** 2 fair
**Contribution:** 2 fair
**Rating:** 5
**Confidence:** 2

**Summary:**

This paper proposes a loss function named Output Contrastive Loss (OCL), which aims to improve capability of models on test-time training for semantic segmentation. The authors find the traditional contrastive loss is not suitable for test-time training of semantic segmentation. Therefore, they enhance the traditional contrastive loss with using a high temperature, using feature from the output space and defining the positive and negative sets based on pixels.  The BN Statistics Modulation and Stochastic Restoration techniques are introduced for higher performance.

**Strengths:**

(1) The proposed method demonstrates better performance than other methods.

(2) The idea is logical and easy to understand.

**Weaknesses:**

(1) Lack of necessary ablation experiments. The authors propose a new loss function called Output Contrastive Loss, and adopt the other two existing techniques BN Statistics Modulation and Stochastic Restoration. Therefore, the key point of ablation experiments should be validating the effectiveness of each component in OCL. The existing experiment results in table 5 show that all of OCL, BN Statistics Modulation and Stochastic Restoration can improve the performance. While the authors should conduct experiments focusing on OCL itself. That is, conduct ablation experiments of using prediction pi/representation zi, defining positive and negative sets according to pixels/images, using high/low temperature.

(2) The idea is not novel enough. Through I'm not familiar with test-time training, I still reckon that the idea of OCL is lack of novelty. The main idea of OCL is employing a high temperature, changing from using feature before head to using feature after head and changing from defining positive and negative sets based on images to pixels. From my point of view, through the three points sound logical, they are not combined well. In other words, they are more like three independent methods. Besides, using positive and negative sets based on pixels is not novel, which has been proposed in previous studies in contrastive learning (for example, PC$^{2}$Seg[1]). For the other two points, the reason of using feature after head is a bit trivial for me. And the reason of defining positive and negative sets based on pixels is not explained well. The authors could provide more experiment results and analyses to prove their importance and provide more insights.

[1] Zhong, Yuanyi, et al. "Pixel contrastive-consistent semi-supervised semantic segmentation." Proceedings of the IEEE/CVF International Conference on Computer Vision. 2021.

**Questions:**

(1) As the ablation results shown in table 5, the performance gains are mainly brought by BN statistic Modulation and stochastic restoration in OCL, which are also adopted by CoTTA. In the table 3 (CS→ACDC), we can see that CoTTA achieves similar performance with OCL. While in the table 2, the CoTTA even decreases the mIou (GTA→CS) when OCL demonstrates much better performance. What's the main difference between the two tasks or datasets that makes CoTTA can not work well? Why does the OCL show much better performance than CoTTA in table 2 compared to table 3?

---

### Official Review · Reviewer_GWxQ · 2023-11-02

**Soundness:** 2 fair
**Presentation:** 2 fair
**Contribution:** 1 poor
**Rating:** 3
**Confidence:** 4

**Summary:**

This paper works on the semantic segmentation in the continual learning set. Specifically, it proposes to run testing time training by the designed contrastive learning in the output space. The derived method is simple yet effective in online adaptation. With well-integrated framework, this work achieves high performance on several settings.

**Strengths:**

1. The writing is straightforward and clear to understand. It clearly explains why executing on the output space and what are the advantages over previous methods.

2. The proposed method is simple, easy to follow and implement. With limited add-on complexity, it proves to improve the generalization ability of the trained networks.

3. Experiments are through and cover lots of scenarios.

**Weaknesses:**

1. There are significant mathematical errors:
- On the left of Fig 1, it is written "a small temperature value ... ensure the uniformity of feature distributions." However, that statement is just the opposite of the truth. Instead, high temperatures will lead to a uniform distribution.
- In the 5th line of Eq 10 (in Supp A), the numerator should be $1 + sim(p_i, p_k) / \tau$ instead of $sim ((1 + p_i, p_k) / \tau)$

2. The derived Eq 3 itself is already intuitive and easy to follow. Deliberately adding Eq 2 and derivations in Supp A is superfluous. Given this point, the main novelty of this paper is limited. BN Modulation and Stochastic Restoration are just lent from previous works.

3. The experiments, though various, are not solid enough.
- In Table 2, OCL only contributes marginally to DAFormer.
- In Table 3, the advantage over CoTTA is limited, which may be from randomness.
- Table 5 shows, OCL brings less improvements than BN. and S.R.

**Questions:**

No questions

---

### Official Review · Reviewer_sv9T · 2023-11-03

**Soundness:** 3 good
**Presentation:** 2 fair
**Contribution:** 3 good
**Rating:** 6
**Confidence:** 3

**Summary:**

This paper proposes a method for test-time training for semantic segmentation in the case of domain adaptation. The proposed method applies contrastive learning on outputs instead of features to improve consistency between the backbone and the head. A high temperature is selected to prevent uniform class distribution. Other tricks such as adapting BN and resetting model parameters are also applied and discussed.

**Strengths:**

1. This paper presents a nice unified solution of the positive term and negative term as the standard infoNCE loss with high temperature.
2. The results are studied and verified in different settings such as TTT and continual-TTT and compared against TTT and DA baselines.
3. Ablations are studied well in various settings and datasets.

**Weaknesses:**

1. Marginal performance gain in the case of DA+TTT. The proposed OCL framework requires training with previous test samples in the test set instead of treating each test sample independently. This is already taking into account almost the full test distribution and thus should be compared with domain adaptation methods. From table 2, DAformer performs significantly better than TTT methods but OCL is improving marginally on top.
2. Limited novelty given the extensive literature on different combination of TTT, contrastive learning, and semantic segmentation. There isn’t sufficient support (or experimental proof) for switching from feature contrast to output contrast, while the temperature and BN stats are mostly minor tricks.
3. Clarification on the setting definition, similar to the Table 1 of TENT and Table 1 of COTTA. If I am understanding correctly, the proposed OCL method does not require pretraining with the OCL loss since OCL works on the output, different from existing TTT methods with SSL aux tasks, but assume online-TTT which uses parameters learned from previous test samples. Given the literature on TTT, online-TTT, Domain Generalization, Domain Adaptation, TTA, fully TTA, etc. such clarification will help readers understand the contribution.
4. Missing clarification in Eq. 2 if the p_i and p_j prediction refers to the prediction after softmax, i.e. they are all positive. Then these are already really similar vectors in the high dimensional space by cosine metric, which caused the uniform class distribution problem.

**Questions:**

It will be interesting to discuss or unify also the output contrastive learning approach with existing pseudo labeling approaches (either in the output space or with a prototype in the feature space).